# Impact of Targeted Agents on Survival of Chronic Lymphocytic Leukemia Patients Fit for Fludarabine, Cyclophosphamide, and Rituximab (FCR) Relative to Age- and Sex-Matched Population

**DOI:** 10.3390/cancers16061085

**Published:** 2024-03-07

**Authors:** Stefano Molica, Tait D. Shanafelt, David Allsup, Diana Giannarelli

**Affiliations:** 1Department Hematology, Hull University Teaching Hospitals NHS Trust, Hull HU3 2JZ, UK; david.allsup@hyms.ac.uk; 2Division of Hematology, Department of Medicine, Stanford University School of Medicine, Stanford, CA 94305, USA; tshana@stanford.edu; 3Centre for Biomedicine, Hull York Medical School, Hull HU6 7RU, UK; 4Biostatistics Unit, Scientific Directorate Fondazione Policlinico Universitario Agostino Gemelli IRCCS, 00168 Roma, Italy; diana.giannarelli@policlinicogemelli.it

**Keywords:** CLL pts fit for FCR, targeted agents, BTKis, venetoclax-based therapy, life-expectancy

## Abstract

**Simple Summary:**

In a comprehensive analysis of phase 3 clinical trials, including the two FLAIR sub-studies, ECOG1912, and the CLL13 trials, we assessed the impact of first-line treatments with targeted agents (TAs), or fludarabine, cyclophosphamide, and rituximab (FCR)-based chemo-immunotherapy (CIT), on overall survival (OS) compared to age- and sex-matched individuals in the general population. TAs demonstrated a higher 5-year restricted mean survival time (RMST) (58.1 months; 95% CI: 57.4 to 58.8) compared to CIT (5-year RMST, 56.9 months; 95% CI: 56.7–58.2). Moreover, the comparison with age- and gender-matched general populations (AGMGP) suggested that TAs may mitigate CLL’s impact on OS during the first five years post-treatment initiation. In contrast, CLL patients treated with FCR exhibited sustained OS differences compared to both the Italian and US AGMGP cohorts. These results support TAs as the preferred first-line treatment for younger/fit CLL patients but imply the need for a careful interpretation due to variations in patient selection criteria and clinical profiles across trials. Longer follow-up is essential to assess the survival improvement of younger CLL patients treated with TAs relative to the AGMGP.

**Abstract:**

To assess the impact of first-line treatment with targeted agents (TAs) or fludarabine, cyclophosphamide, and rituximab (FCR)-based chemo-immunotherapy (CIT) on overall survival (OS) compared to age- and sex-matched individuals in the general population, we conducted an aggregated analysis of phase 3 clinical trials, including the two FLAIR sub-studies, ECOG1912, and CLL13 trials. The restricted mean survival time (RMST), an alternative measure in outcome analyses capturing OS changes over the entire history of the disease, was used to minimize biases associated with the short follow-up time of trials. Patients treated with TAs demonstrated a higher 5-year RMST (58.1 months; 95% CI: 57.4 to 58.8) compared to those treated with CIT (5-year RMST, 56.9 months; 95% CI: 56.7–58.2). Furthermore, the OS comparison of treatment groups with the AGMGP suggests that TAs may mitigate the impact of CLL on OS during the first five years post-treatment initiation. In summary, the 5-year RMST difference was −0.4 months (95% CI: −0.8 to 0.2; *p* = 0.10) when comparing CLL patients treated with TAs to the Italian age- and gender-matched general population (AGMGP). A similar trend was observed when CLL patients treated with TAs were compared to the US AGMGP (5-year RMST difference, 0.3 months; 95% CI: −0.1 to 0.9; *p* = 0.12). In contrast, CLL patients treated with FCR exhibited sustained OS differences when compared to both the Italian cohort (5-year RMST difference: −1.6 months; 95% CI: −2.4 to −0.9; *p* < 0.0001) and the US AGMGP cohort (5-year RMST difference: −0.9 months; 95% CI: −1.7 to −0.2; *p* = 0.015). Although these results support TAs as the preferred first-line treatment for younger CLL patients, it is crucial to acknowledge that variations in patient selection criteria and clinical profiles across clinical trials necessitate a cautious interpretation of these findings that should be viewed as directional and hypothesis-generating. A longer follow-up is needed to assess the survival improvement of younger CLL patients treated with TAs relative to the AGMGP.

## 1. Introduction

Despite a significant improvement in clinical outcome for individuals with chronic lymphocytic leukemia (CLL) treated with targeted agents (TAs), these therapies are not considered curative [1]. Nevertheless, data derived from three phase 3 trials assessing ibrutinib either as a monotherapy (RESONATE2) or in combination with anti-CD20 therapy (ECOG1912 and iLLUMINATE) in previously untreated patients with CLL/small lymphocytic lymphoma (SLL) suggest that the use of first-generation Bruton’s tyrosine kinase inhibitors (BTKis) may extend life expectancy compared to traditional chemoimmunotherapy (CIT)-based treatments [2]. Furthermore, the same study indicates that the life expectancy of elderly CLL patients treated with ibrutinib approximates that of an age-matched population aged ≥65 years [2]. These findings have only been partially replicated in a comprehensive analysis of first-line treatments, including venetoclax or trials involving second-generation BTKis, which specifically enrolled patients aged 65 years or older [3]. Differences in comorbidity burdens across trials, as well as between clinical trial participants and individuals within the AGMGP cohorts, could have influenced the study outcomes [3].

Less is known about the impact of first-line treatment with TAs on the survival of younger CLL patients. Several phase 3 clinical studies have compared ibrutinib-based combination or venetoclax-based treatments to treatment with fludarabine, cyclophosphamide, and rituximab (FCR) in younger, fit, CLL patients. These trials have indicated that TAs offer superior progression-free survival (PFS) relative to CIT [4,5,6,7,8]. The 5-year results of the US ECOG1912 trial demonstrated an OS benefit for patients receiving continuous ibrutinib combination therapy over those treated with FCR [4,5], although the difference in OS was not statistically significant in the similarly designed FLAIR trial [7]. However, robust data on the true impact of TAs on OS in younger CLL patients are lacking due to the relatively short follow-up of phase 3 studies for a disease like CLL, which has a protracted natural history.

To assess the impact of first-line treatment with TAs and FCR-based CIT on OS in comparison to age- and sex-matched individuals in the general population, we conducted an aggregated analysis of phase 3 clinical trials. The restricted mean survival time (RMST), an alternative measure in outcome analyses that captures the OS changes over the entire history of disease, was used. Of note, this approach mitigates biases linked to the limited follow-up duration often encountered in clinical trials [9].

## 2. Materials and Methods

### 2.1. Search Strategies, Studies’ Eligibility, and Data Extraction

A comprehensive search of the MEDLINE database was conducted to identify relevant full-text articles and research letters reporting the results of phase 3 TA clinical trials enrolling younger/fit, treatment-naïve patients with CLL. The search strategy utilized both subject headings (MESH) terms (e.g., chronic lymphocytic leukemia, targeted agents, younger/fit patients, FCR) and free-text words to enhance sensitivity.

Studies were included based on specific eligibility criteria, with a prerequisite being the presence of an FCR CIT comparator arm and an experimental arm investigating inhibitors targeting the Bruton’s tyrosine kinase (BTK) or B-cell lymphoma 2 (BCL2) cell signaling pathways. Given the focus of analysis, only trials providing information on OS were included. The reporting adhered to the Preferred Reporting Items for Systematic Reviews and Meta-Analyses (PRISMA) statement, serving as a robust reporting guideline (Figure 1) [10].

Abstract evaluation and data extraction were performed by two reviewers (S.M. and D.G.) with a cross-audit between reviewers. Data, extracted in a standardized format, included patient characteristics, study characteristics, *IGHV* status, *TP53* status, results of chromosome analysis by interphase fluorescence in situ hybridization (FISH) (i.e., 11q deletion, 17p deletion), treatment regimen, duration of follow-up, PFS, and OS.

### 2.2. Statistical Analyses

Analysis of individual patient data (IPD) is the gold standard approach for secondary analyses of survival. This approach, however, requires access to IPD from all studies. An alternative method to reconstruct this information from the number of patients at risk and Kaplan–Meier (KM) curves in published studies was introduced by Liu et al. [11]. This approach has been validated in several studies [12,13]. The two stage method for data reconstruction, publicly accessible online at https://www.trialdesign.org/one-page-shell.html#IPDfromKM (21 January 2024), was used for the present study. In the first stage, quality data coordinates from KM curves, including survival probability and time, were extracted using a digital technique. This technique involves providing the number of patients at risk at different time points. In the second stage, the algorithm meticulously checks the consistency of KM steps with both the occurrence of events and the number of censored patients in specific time intervals. The accuracy of reconstruction is further ensured by iteratively verifying the overlap with the original plot until achieving an exact reproduction of the figure reported in the paper [11].

The RMST was subsequently utilized to estimate the average survival time over a fixed 5-year period [9]. The RMST is defined as the area under the survival curve up to a specific time point and is generally more reliably estimable than mean or median survival times. In our analysis, we selected the 5-year time point because it corresponds to the median follow-up time of most included studies [4,6,7,8]. This time point was explicitly chosen to obtain an RMST, reflecting the clinically relevant time horizon. Importantly, RMST serves as a statistical measure particularly useful in addressing biases associated with differences in follow-up lengths in survival analysis, especially within the context of clinical trials or observational studies. For these reasons, RMST is proposed as a novel alternative measure in survival analyses and may prove beneficial when the proportional hazards assumption cannot be made or when the event rate is low [9].

Both Italian and US age- and gender-matched general population (AGMGP) were used as control populations due to differences in life expectancy between the general populations of the United States and Italy [14,15]. The expected OS for the age and sex matched population in these two countries was determined using data from the 2019 Centers for Disease Control and Prevention (CDC) life table and the 2019 Italian Istituto Nazionale di Statistica (ISTAT). It is noteworthy that these pre-pandemic databases remained unaffected by biases related to COVID-19 mortality [14,15].

The RMST difference, which compares RMSTs between CLL patients initiating first-line treatment and the AGMGP, was used as an indicator of the treatment effect and to provide insights into the magnitude to which specific treatments can mitigate the adverse impact of CLL on OS. Negative RMST values indicate a shorter survival of CLL patients compared to the Italian or US population [9].

## 3. Results

### 3.1. Studies Selected and Patient Characteristics

The search approach previously detailed (see Methods) led to the identification of four phase 3 studies: the two FLAIR sub-studies, ECOG1912, and the CLL13 trials [4,6,7,8]. FLAIR is an adaptive design, initially comparing ibrutinib and rituximab (IR) to FCR in previously untreated CLL patients. In 2017, the trial was modified to include ibrutinib monotherapy and ibrutinib combined with venetoclax (IV), with therapy duration determined by minimal residual disease (MRD) criteria. Accordingly, patients were randomly assigned in a 1:1:1 ratio to receive FCR, ibrutinib monotherapy, or IV. For this analysis, we focused on patients enrolled in the initial FLAIR sub-study comparing FCR to IR and the subsequent FLAIR sub-study comparing FCR to IV. Also, patients included in the CLL13 trial comparing CIT (FCR or bendamustine-rituximab [BR]) to venetoclax + rituximab [VR] or venetoclax + obinutuzumab [VO] or VO + ibrutinib [VOI]) and those enrolled in the ECOG 1912 study comparing IR in a 2:1 ratio to FCR were included [4,6,7,8] (Table 1).

Among the 2751 patients enrolled in these trials, 1697 (61.7%) received treatment with TAs and 1054 (38.2%) received CIT (975 or 92.5% FCR, and 79 or 7.4% BR; Figure 2). Because the outcomes of patients receiving FCR or BR in CLL13 trial were presented in aggregate form, we were unable to exclude the latter in our survival analysis.

Despite slight variations in age eligibility criteria across trials, all enrolled patients were deemed suitable candidates for FCR therapy. The FLAIR trial set the upper age limit for patient enrollment at 75 years, and ECOG1912 set the upper age limit at 70 years. In the CLL13 trial, a 65-year threshold identified patients as being suitable for either FCR or BR.

In the aggregate analysis, median age was 61.5 years (range: 27 to 84) and the median proportion of male patients was 72% (range: 67.3–74). Among patients with available genetic data, 1448 out of 2749 (52.6%) had unmutated *IGHV*, 493 out of 274 (17.9%) had 11q(del), and 4 out of 2749 (0.001%) had del(17p)/mutated TP53. Median follow-up was 38.8, 53, 43.7, and 68 months for CLL13, FLAIR, adapted FLAIR, and ECOG 1912, respectively [4,5,6,7,8] (Table 2).

### 3.2. OS of CLL Patients Treated with TAs or FCR Compared to AGMGP

After pooling patient data from ECOG1912, the two FLAIR sub-studies, and the CLL13 trials, we conducted a comprehensive analysis to evaluate the impact of TAs and CIT on OS of CLL patients. In this comparative analysis, we observed that the 5-year RMST was longer for patients treated with TAs (58.1 months; 95% CI: 57.4 to 58.8) compared to those treated with CIT (5-year RMST, 56.9 months; 95% CI: 56.7–58.2) (Figure 3A,B).

Furthermore, our findings suggest that only TAs demonstrated the ability to mitigate the impact of CLL on OS during the initial five years post-treatment initiation. The 5-year RMST difference comparing CLL patients treated with TAs relative to the Italian AGMGP was −0.4 months (95% CI: −0.8 to 0.2; *p* = 0.10). A similar trend was observed in the comparison between CLL patients treated with TAs and the US AGMGP (5-year RMST difference, 0.3 months; 95% CI: −0.1 to 0.9; *p* = 0.12; see Figure 3C).

In contrast, a statistically significant RMST difference was noted when comparing the OS estimates of the FCR patient cohort with both the Italian (5-year RMST difference: −1.6 months; 95% CI: −2.4 to −0.9; *p* < 0.0001) and US AGMGP (5-year RMST difference: −0.9 months; 95% CI: −1.7 to −0.2; *p* = 0.015; Figure 3C).

### 3.3. OS of Patients Treated with Different TAs Compared to the AGMGP

In subsequent analyses, we investigated the impact of different TAs on OS of CLL patients compared to the AGMGP. For this analysis, patients were categorized into the following two treatment groups:

(i) Patients treated with indefinite ibrutinib-based regimens (IR; n = 740, 43.6%), comprising those included in the experimental arms of the ECOG1912 trial and the initial FLAIR sub-study [4,7].

(ii) Patients treated with fixed-duration (FD) venetoclax-based combinations, including those enrolled in the experimental arms of CLL13 (VO, [n = 229, 13.5%]; VR, [n = 237, 13.9%]; VOI, [n = 231, 13.6%]) and in the more recently published FLAIR sub-study (VI, [n = 260, 16.3%]; Figure 1) [6,8].

Comparable OS estimates were observed for patients treated with either indefinite ibrutinib regimens or FD venetoclax-based therapies. The 5-year RMST values were 58 months (95% CI: 57.3 to 58.7) for indefinite ibrutinib regimens and 58.1 months (95% CI: 57.4 to 58.8) for FD venetoclax-based therapies (Figure 4A,B).

In the OS comparison to the Italian AGMGP, both indefinite ibrutinib regimens and FD venetoclax-based therapies appeared to mitigate the negative impact of CLL on OS. The 5-year RMST difference was −0.5 months (95% CI: −1.2 to 0.2; *p* = 0.08) in comparison with patients treated with continuous ibrutinib and 0.4 months (95% CI: −1.0 to 0.2; *p* = 0.10) in comparison with those treated with venetoclax-based combinations (Figure 4C).

Similar results were observed when comparing OS of the US AGMGP to OS of CLL patients treated with either indefinite ibrutinib regimens (5-year RMST, 0.2 months; 95% CI: −0.5 to 0.9; *p* = 0.29) or venetoclax-based combinations (5-year RMST, 0.3 months; 95% CI: −0.3 to 0.9; *p* = 0.16) (Figure 4C).

## 4. Discussion

Despite the widespread use of TAs in the treatment landscape of CLL, the impact of TAs on OS remains unclear. Our aggregate analysis evaluating the absolute survival differences in TAs relative to FCR-based CIT and the 5-year OS estimates of CLL patients treated with TAs relative to the AGMGP suggests that TAs mitigate the negative impact of CLL on OS over the first 5 years from treatment initiation. In contrast, a statistically significant shorter OS at 5 years relative to the AGMGP persists for CLL patients treated with FCR-based CIT. Given the imperative to maximize therapeutic benefits while minimizing long-term side effects, our results provide additional evidence that TAs outperform FCR in terms not only for PFS but also OS [16].

Our analysis is subject to several limitations. First, the algorithm used for individual patient data reconstruction lacked some variables, such as race and ethnicity, as well as some individual CLL risk characteristics. While providing a close approximation of patient-level survival, the absence of these variables limits the depth of our analysis [10]. Second, although the results strongly support TAs as the preferred first-line treatment for younger/fit CLL patients, longer follow-up is needed to assess the relative survival of fit CLL patients treated with TA relative to the general population. Indeed, the follow-up duration falls short for a comprehensive assessment of relative survival among younger CLL patients compared to the general population, necessitating extended follow-up periods for a more profound understanding. From a methodological standpoint the use of RMST, generally preferred to classic hazard ratio when the event rate is very low or the follow-up time is very short, only partially overcomes this limitation [17]. Finally, it should be noted that all trials involving BTK inhibitors identified by our search used the first-generation BTK inhibitor ibrutinib [4,5,6,7,8].

Our comprehensive study indicates that the enhanced survival rates observed among younger and fit CLL patients, as highlighted in this aggregated analysis, hold promise for mitigating the adverse impact of CLL on overall life expectancy. It is imperative to acknowledge, however, that variations in patient selection criteria and clinical profiles across clinical trials necessitate a cautious interpretation of these findings. Rather than serving as conclusive evidence, our results should be regarded as a directional guide and a hypothesis-generating foundation. This nuanced perspective underscores the complexity of CLL management, emphasizing the need for continued investigation to refine treatment strategies and tailor interventions to specific patient profiles.

## 5. Conclusions

In summary, our comprehensive investigation reveals that the administration of TAs yields a favorable impact on OS within the subset of younger and physically fit CLL patients. Notably, this positive outcome is particularly pronounced when compared with the conventional FCR-based CIT. The discernible enhancement in survival rates serves as a crucial advancement in ameliorating, albeit partially, the adverse impact of CLL on the life expectancy of affected individuals.

As we navigate the ever-evolving landscape of CLL treatment, ongoing advancements in TA strategies have the potential to further elevate the efficacy of TA approaches. Noteworthy among these are the incorporation of second-generation BTK inhibitors either as standalone therapies or in synergistic combinations with BCL2 inhibitors. Concurrently, the exploration of treatment duration guided by measurable residual disease (MRD) adds a layer of precision to the therapeutic landscape [18,19,20,21].

Looking ahead, an imperative facet of our study underscores the necessity for extended follow-up assessments in ongoing and future clinical trials. This longitudinal scrutiny becomes paramount to accurately gauge the relative survival improvements experienced by physically fit CLL patients treated with TAs when compared to the general population. This ongoing evaluation will be pivotal in discerning the sustained efficacy and long-term benefits of TAs, thus contributing significantly to the evolving landscape of CLL management.

## Figures and Tables

**Figure 1 cancers-16-01085-f001:**
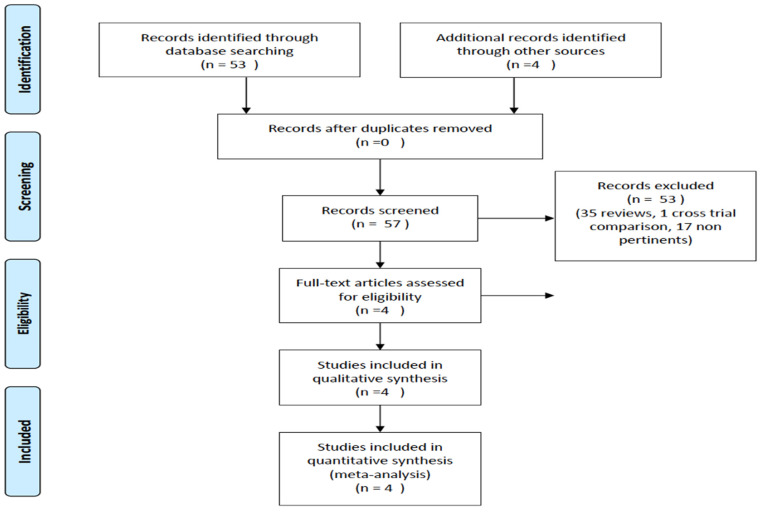
PRISMA diagram. https://guides.lib.unc.edu/prisma (access on 21 January 2024).

**Figure 2 cancers-16-01085-f002:**
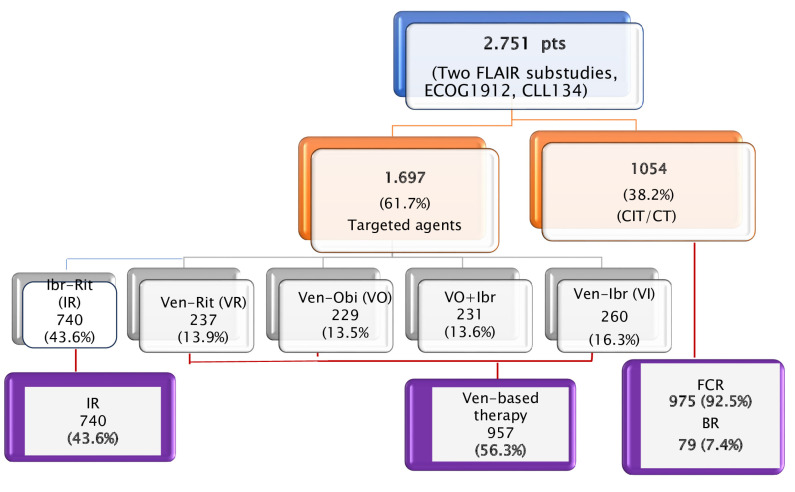
Randomized clinical trials enrolling treatment-naïve chronic lymphocytic leukemia (CLL) patients fit for fludarabine, cyclophosphamide, rituximab (FCR)-based chemoimmunotherapy [4,6,7,8]. For further analyses, patients were merged into three treatment groups: (i) indefinite ibrutinib-based regimens (i.e., ibrutinib-rituximab [IR]); (ii) fixed-duration (FD) venetoclax-based combinations (venetoclax-obinutuzumab [VO], venetoclax-rituximab [VR], VO + ibrutinib [VOI], and venetoclax-ibrutinib [VI]); (iii) chemo-immunotherapy.

**Figure 3 cancers-16-01085-f003:**
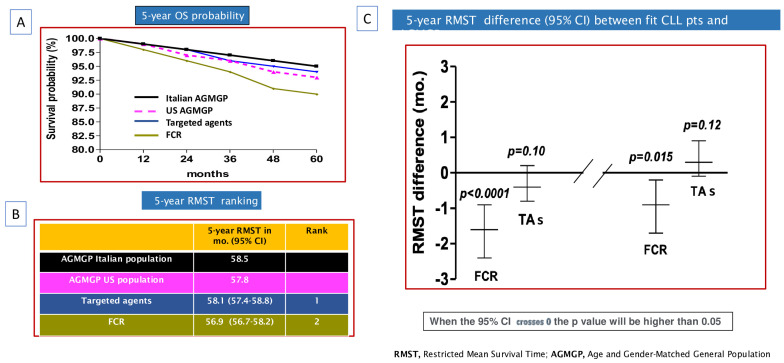
Overall survival (OS) of chronic lymphocytic leukemia (CLL) patients included in phase 3 clinical trials (two FLAIR sub-studies, CLL13, and ECOG1912), Italian age- and gender-matched general population (AGMGP) and the US AGMGP: (**A**) OS Kaplan–Meier curves comparing CLL patients enrolled in the TA arms or FCR-based CIT arm of first-line therapy trials to US AGMGP and Italian AGMGP. (**B**) Five-year restricted mean survival time (RMST) ranking of CLL patients, Italian AGMGP, and US AGMGP. (**C**) Five-year RMST difference comparing CLL patients, Italian AGMGP, and US AGMGP. In all analyses, patients were grouped into one of two treatment categories: (i) pooled targeted agent (TA) group; (ii) FCR-based chemoimmunotherapy (CIT) group.

**Figure 4 cancers-16-01085-f004:**
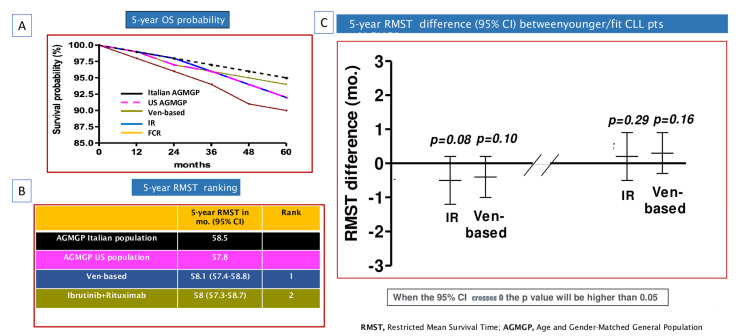
Overall survival (OS) comparison of chronic lymphocytic leukemia (CLL) patients included in phase 3 clinical trials (two FLAIR sub-studies, CLL13, and ECOG1912), Italian age- and gender-matched general population (AGMGP), and the US AGMGP: (**A**) OS Kaplan–Meier curves comparing CLL patients enrolled in TA arms of the trials to US AGMGP and Italian AGMGP (**B**) Five-year restricted mean survival time (RMST) ranking of CLL patients, Italian AGMGP, and US AGMGP. (**C**) Five-year RMST difference comparing CLL patients to Italian AGMGP and US AGMGP. In these analyses CLL patients were grouped into one of two treatment categories: (i) indefinite ibrutinib + rituximab (IR); (ii) fixed-duration venetoclax-based (Ven-based) combinations. The 95% confidence interval (CI) crossing zero indicates that the *p* value was greater than 0.05.

**Table 1 cancers-16-01085-t001:** Design of the phase 3 clinical trials included in the present analysis.

	Type of Study	Recruiting from	Inclusion Criteria	Experimental Arm	Control Arm	Randomization Ratio
1° FLAIR sub-study [7]	RCT	UK Institutions	Pts. fit for FCR	IR	FCR	1:1
2° FLAIR sub-study [8]	RCT	UK Institutions	Pts. fit for FCR	IR or I	FCR	1:1:1
ECOG1912 [4,5]	RCT	ECOG Centers	Pts. fit for FCR	IR	FCR	2:1
CLL13 [6]	RCT	GCLLSG/HOVON centers	CIRS score < 6/Cr Cl > 70	VR/VO/VOI	FCR *	1:1:1:1

RCT, randomized clinical trial; UK, United Kingdom; FCR, fludarabine, cyclophosphamide, rituximab; IR, ibrutinib/rituximab; I, ibrutinib; ECOG, Eastern Cooperative Oncology Group; GCLLSG, German CLL Study Group; HOVON, Hemato-Oncology Foundation for Adults in the Netherlands; VR, venetoclax/rituximab; VO, venetoclax/obinutuzumab; VOI, venetoclax/obinutuzumab/ibrutinib; CIRS, cumulative illness rating Scale. (*) 7.4% of patients received bendamustine/rituximab (BR).

**Table 2 cancers-16-01085-t002:** Characteristics of patients enrolled in the two FLAIR sub-studies, or in ECOG1912, or the CLL13 trials.

Study	Experimental Arm	No Pts. in Experimental Arm	Pts Median Age in Years, (Range)	Male	Median Follow-Up (Mo)	Pts. with *U-IGHV*	Pts. with 11q (Del)	Pts with *TP53* Aberrations
First FLAIR sub-study [7]	Ibrutinib + Rituximab	386	63 (55–67)	73.0%	53.0	50.0%	15%	<1%
CLL13 [6]	VR VO VO + Ibrutinib	237 229 231	62 (27–84) 62 (31–83) 60 (30–84)	73.8% 74.7% 68.4%	38.8	56.5% 57.0% 53.2%	19.0% 19.2% 13.9%	NR
ECOG1912 [4,5]	Ibrutinib + Rituximab	354	56.7 (±7.5) *	62.0%	69.7	75.0%	22.0 %	<1%
Second FLAIR sub-study [8]	Ibrutinib +Venetoclax	260	62 (56–67)	71.3%	43.7	49.9%	18.2%	<1%

VR: venetoclax + rituximab; VO: venetoclax + obinutuzumab; NR, not reported. * In the ECOG1912 trial, age is provided as mean rather than median.

## Data Availability

Data available on request from the authors.

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
