# Peer review of "Impact of Targeted Agents on Survival of Chronic Lymphocytic Leukemia Patients Fit for Fludarabine, Cyclophosphamide, and Rituximab (FCR) Relative to Age- and Sex-Matched Population"

_cancers, 2024, doi:10.3390/cancers16061085_

Round 1

Reviewer 1 Report

Comments and Suggestions for Authors

This is a comprehensive and interesting paper by Molica and colleagues regarding the increasingly central. role of target therapy in CLL. I have only minor suggestions regarding Figures and Tables, where some parts of the text are cut (Fig 3C) or could be aesthetically improved (Fig.2, Table 1)

Author Response

This is a comprehensive and interesting paper by Molica and colleagues regarding the increasingly central. role of target therapy in CLL. I have only minor suggestions regarding Figures and Tables, where some parts of the text are cut (Fig 3C) or could be aesthetically improved (Fig.2, Table 1)

Dear referee,

We would like to thank you for these favorable comments.

We appreciate the suggestions to improve the quality of tables/figures and we paid now particular attention to including in text the cut parts.

Reviewer 2 Report

Comments and Suggestions for Authors

Authors aimed to assess the impact of first-line treatment with targeted agents (TAs) and fludarabine, cyclophosphamide, and rituximab (FCR)-based chemo-immunotherapy (CIT) on overall survival (OS) compared to age- and sex-matched individuals in the general population, and they conducted an aggregated analysis of phase 3 clinical trials, including the two FLAIR sub-studies, ECOG1912, and CLL13 trials.

In my opinion, this study has many lacks as they also mentioned in Discussion part and it is not well-established and well-presented. Therefore, I recommend reject for publication in Cancers journal even in a communication form.

Here are the points:

·      In Introduction, Bruton’s Tyrosine Kinase (BTK) and B-Cell Lymphoma 2 (BCL2), the most important targets of CLL and the drugs affecting these targets should be written in detail.

·      In Discussion, authors mentioned the lacks of this study in detail, which this study cannot be acceptable.

·      There are second and third generation of BTK inhibitors, with superior activity and less side effects compared to ibrutinib. The selectivity of ibrutinib on BTK is poor, it also affects other kinases, and there are severe side effects with ibrutinib.

·      It is not clear what the phrase “fit CLL patients”. What is the definition of being fit in the study?

·      The study is age- and gender-matched general populations but there are two different ethnic groups.

·      The characteristics of patients are not mentioned in detail. How about the other factors of each patient, such as use of other drugs, any other disease to affect the study?

·      There are too many short terms of phrases that makes the paper to understand impossible such as TA, AGMGP…

·      They should replace https://www.trialdesign.org/one-page-shell.html#IPDfromKM to the reference part.

Author Response

Dear Referee,

In the revised manuscript we are submitting, we tried to satisfy all points raised.

Authors aimed to assess the impact of first-line treatment with targeted agents (TAs) and fludarabine, cyclophosphamide, and rituximab (FCR)-based chemo-immunotherapy (CIT) on overall survival (OS) compared to age- and sex-matched individuals in the general population, and they conducted an aggregated analysis of phase 3 clinical trials, including the two FLAIR sub-studies, ECOG1912, and CLL13 trials.

In my opinion, this study has many lacks as they also mentioned in Discussion part and it is not well-established and well-presented. Therefore, I recommend reject for publication in Cancers journal even in a communication form.

Here are the points:

  • In Introduction, Bruton’s Tyrosine Kinase (BTK) and B-Cell Lymphoma 2 (BCL2), the most important targets of CLL and the drugs affecting these targets should be written in detail.

At least in the authors’ opinion, there is no need to further discuss in this paper “Bruton’s Tyrosine Kinase   (BTK) and B-Cell Lymphoma 2 (BCL2), the most important targets of CLL and the drugs affecting these targets”. The aim of this paper is mainly to explore the potential of novel therapy to mitigate the negative impact of CLL on OS.

  • In Discussion, authors mentioned the lacks of this study in detail, which this study cannot be acceptable.

While novel agents have improved the outcome of CLL patients in terms of prolonged PFS what is not clear is their impact on OS. In particular, especially in the context of younger/fit CLL patients (those eligible for FCR therapy) it is not clear whether novel agents have mitigated the impact of CLL on OS estimates. We addressed, for the first time, this issue.

  • There are second and third generation of BTK inhibitors, with superior activity and less side effects compared to ibrutinib. The selectivity of ibrutinib on BTK is poor, it also affects other kinases, and there are severe side effects with ibrutinib.

We agree that are now available second and third generation BTK inhibitors. While the patient safety is improved with these novel agents it is debatable whether they  are more  efficactive than ibrutinib, the first-in-class BTKi.

However, second or third generation BTKis were never tested in phase 3 clinical trials having as control arm FCR.

  • It is not clear what the phrase “fit CLL patients”. What is the definition of being fit in the study?

This definition was better clarified in this revised version. We clearly refer to fit/younger patients, in other words those patients eligible for FCR.

  • The study is age- and gender-matched general populations but there are two different ethnic groups.

I agree, only in part. Please consider that 3 out of 4 studies were conducted in Europe (the two FLAIR studies was conducted in UK and CLL3 in Germany and Netherland). Only the ECOG1912 was conducted in US.

  •  

The characteristics of patients are not mentioned in detail. How about the other factors of each patient, such as use of other drugs, any other disease to affect the study?

In table 2 are included the clinic-biological characteristics of patients included in the present analysis.

Overall it seems that they are homogeneously distributed across trials.

The impact of comorbidities or polypharmacy is really limited among patients enrolled in these trails trials.  We are dealing with studies enrolling younger CLL patients fit for FCR.

  •  

There are too many short terms of phrases that makes the paper to understand impossible such as TA, AGMGP…

We tried to avoid an extensive use acronyms .

  •  

 They should replace https://www.trialdesign.org/one-page-shell.html#IPDfromKM to the reference part.

It was done

Reviewer 3 Report

Comments and Suggestions for Authors

This study performed a meta-analysis using restricted mean survival time for survival analysis. RMST is a novel alternative analysis of overall survival that measures the area under the curve up to a specific time point. The data indicated that TA treatment in CLL patients is better than FRC, and with TA the patients have a survival similar to the rest of the population. The manuscript is well written, it is easy to read and understand.

Comments:

(1) Please note that the abstracts shown in the pdf file and the mdpi webpage are different.

(2) Line 114. Please add accessed on..."specific date" after the webpage.

(3) Could you please explain why 5-year period, the 5-year time point was selected?

(4) Line 122. You may expand the description of RMST for survival analysis.

For example:

• Restricted mean survival time (RMST) is suggested as a novel alternative measure in survival analyses and may be useful when proportional hazards assumption cannot be made or when event rate is low.
• RMST is defined as the area under the survival curve up to a specific time point and is generally more reliably estimable than mean or median survival times.
• The time point should be explicitly chosen to obtain an RMST to reflect the clinically relevant time horizon.
• In the case of crossing survival curves, the efficacy of an intervention may be demonstrated by showing a difference in RMST between two curves although the log-rank test may fail to detect differences.

doi: 10.3348/kjr.2022.0061

(5) Line 133. During covid-19 pandemic, has the human mortality increased significantly in comparison to other years with "conventional" influenza?

(6) Regarding the matched population. Why are you comparing with age and gender-matched general population? Should you not compare the two types of treatment? FCR CIT vs. BTK or BCL2 signaling inhibitors?

(7) Lines 142-153. Is it possible to make a table showing the 3 design of the 3 types of study?

(8) As I understand from Table 1. All the clinicopathological characteristics of the patients were comparable between the 4 studies. Therefore, it is correct/adequate to merge all the patients in one analysis as shown in Figure 1.

(9) In figure 3a, I understand that the k-m with log rank test was not statistically significant. Is it right?

(10) When you compare difference of RMST and ratio of RMST between groups, you calculate statistical difference, don't you? What was the difference between FCR vs. TAs in Figure 3?

(11) In Figure 4a. Why italian line is dashed in black?

(12) Since this is a metaanalysis, why a conventional meta-analysis with forest plot was not performed?

(13) Apart from OS, what about other clinical variables such as disease progression, clinical response, etc?

Author Response

Dear Referee,

I would like to thank you for the constructive review of the paper and important suggestions.

This revised version deeply considers your advice and relevant changes are included.

This study performed a meta-analysis using restricted mean survival time for survival analysis. RMST is a novel alternative analysis of overall survival that measures the area under the curve up to a specific time point. The data indicated that TA treatment in CLL patients is better than FRC, and with TA the patients have a survival similar to the rest of the population. The manuscript is well written, it is easy to read and understand.

Thank you for these comments and suggestions that significantly contributed to improve the paper.

Comments:

  • Please note that the abstracts shown in the pdf file and the mdpi webpage are different.

Thank you, please refer to abstracts of this revised version

  • Line 114. Please add accessed on..."specific date" after the webpage.

The specific date was now added

  • Could you please explain why 5-year period, the 5-year time point was selected?

We clarified the reason why 5-year period was selected as time-point. In brief, it corresponds to median follow-up time of most studies selected for this analysis.

  • Line 122. You may expand the description of RMST for survival analysis.

Thank you for these suggestions that contribute to better clarify to the journal’ readers advantages related to the use of restricted mean survival time (RMST).

In the methods we provide now an extensive explanation of RMST.

For example:

  • Restricted mean survival time (RMST) is suggested as a novel alternative measure in survival analyses and may be useful when proportional hazards assumption cannot be made or when event rate is low.
    • RMST is defined as the area under the survival curve up to a specific time point and is generally more reliably estimable than mean or median survival times.
    • The time point should be explicitly chosen to obtain an RMST to reflect the clinically relevant time horizon.
    • In the case of crossing survival curves, the efficacy of an intervention may be demonstrated by showing a difference in RMST between two curves although the log-rank test may fail to detect differences.

doi: 10.3348/kjr.2022.0061

  • Line 133. During covid-19 pandemic, has the human mortality increased significantly in comparison to other years with "conventional" influenza?

We have used data related to 2019 for both Italian and US populations to avoid the bias related to the excess of deaths COVID19-related.. It is believed that COVID19 mortality outperforms the yearly mortality due to conventional epidemic flu.

  • Regarding the matched population. Why are you comparing with age and gender-matched general population? Should you not compare the two types of treatment? FCR CIT vs. BTK or BCL2 signaling inhibitors?

The main aim of our study was to assess the impact of first-line treatment with targeted agents (TAs) and fludarabine, cyclo-phosphamide, and rituximab (FCR)-based chemo-immunotherapy (CIT) on overall survival (OS) compared to age- and sex-matched individuals in the general population.

However, also results regarding a comparison between CIT and targeted agents are provided.  

We clearly argue that the 5-year RMST was significantly higher for  patients treated with targeted agents in comparison to those treated with CIT (58.1 vs 56.9 months).

With regards to the comparison treated respectively with ibrutinib-based and venetoclax-based combinations, no difference in the 5-year RMST was found (58.1 vs 58 months)

  • Lines 142-153. Is it possible to make a table showing the 3 design of the 3 types of study?

A new table (Table 1) including information on the type of studies is now included

  • As I understand from Table 1. All the clinicopathological characteristics of the patients were comparable between the 4 studies. Therefore, it is correct/adequate to merge all the patients in one analysis as shown in Figure 1.

As shown in table 2, the clinico-hematological characteristics of patients were comparable across the 4 studies. For the purpose of this analysis patients were categorized in 3 different groups: those treated with CIT, ibrutinib-based and venetoclax-based combinations, respectively.

This stratification allows for an appropriate comparison in terms of OS with age and sex control populations.

  • In figure 3a, I understand that the k-m with log rank test was not statistically significant. Is it right?

In the figure 3a we provide a KM analysis of different therapeutic group patients (CIT, ibrutinib-based, and venetoclax-based combinations) and Italian or US control populations. These survival curves, presented only visually,  were not compared by log-rank test.

More appropriately, comparison were performed by using the RMST difference (figure 3c and 4c) which is a more suitable method to identify differences between therapeutic groups and control populations.

  • When you compare difference of RMST and ratio of RMST between groups, you calculate statistical difference, don't you? What was the difference between FCR vs. TAs in Figure 3?

The comparison between FCR and TAs is provided in term of RMST. Values are significantly longer for patients treated with TAs in comparison to those treated with FCR (58.1 vs 56.9 months)(Fig 3b)

(11) In Figure 4a. Why italian line is dashed in black?

Both in the figure 3a and 4c KM curve of Italian population is dashed in black to better differentiate visually  it from OS curve of US (dashed in pink) and OS curves  of CLL patients.

(12) Since this is a meta-analysis, why a conventional meta-analysis with forest plot was not performed?

Thank you for this comment. I agree this is a meta-analysis, however, as clearly argued in the method section, the comparative analysis is not based on the hazard analysis comparison.

As extensively explained we used the RMST which better fits with this type of comparative analysis.  

(13) Apart from OS, what about other clinical variables such as disease progression, clinical response, etc?

Thank you for rising this point. The analysis of PFS does not fall in the scope of our analysis. The aim of present study is to explore the potential of novel therapy to mitigate the negative impact of CLL on OS.

Round 2

Reviewer 2 Report

Comments and Suggestions for Authors

Authors revised the manuscript thoughly answering the main questions point by point. Therefore, it can be accepted in the current form as a communication paper in Cancers journal.